# Biowelding 3D-Printed Biodigital Brick of Seashell-Based Biocomposite by *Pleurotus ostreatus* Mycelium

**DOI:** 10.3390/biomimetics8060504

**Published:** 2023-10-23

**Authors:** Yomna K. Abdallah, Alberto T. Estévez

**Affiliations:** iBAG-UIC Barcelona, Institute for Biodigital Architecture & Genetics, Universitat Internacional de Catalunya, 08017 Barcelona, Spain

**Keywords:** bio-welding, seashell-based biocomposite, paste extrusion, 3D-printed brick, elastic brick, mycelium brick, cellulose-based medium, living architecture, *Pleurotus ostreatus*

## Abstract

Mycelium biocomposites are eco-friendly, cheap, easy to produce, and have competitive mechanical properties. However, their integration in the built environment as durable and long-lasting materials is not solved yet. Similarly, biocomposites from recycled food waste such as seashells have been gaining increasing interest recently, thanks to their sustainable impact and richness in calcium carbonate and chitin. The current study tests the mycelium binding effect to bioweld a seashell biocomposite 3D-printed brick. The novelty of this study is the combination of mycelium and a non-agro–based substrate, which is seashells. As well as testing the binding capacity of mycelium in welding the lattice curvilinear form of the V3 linear Brick model (V3-LBM). Thus, the V3-LBM is 3D printed in three separate profiles, each composed of five layers of 1 mm/layer thickness, using seashell biocomposite by paste extrusion and testing it for biowelding with *Pleurotus ostreatus* mycelium to offer a sustainable, ecofriendly, biomineralized brick. The biowelding process investigated the penetration and binding capacity of the mycelium between every two 3D-printed profiles. A cellulose-based culture medium was used to catalyse the mycelium growth. The mycelium biowelding capacity was investigated by SEM microscopy and EDX chemical analysis of three samples from the side corner (S), middle (M), and lateral (L) zones of the biowelded brick. The results revealed that the best biowelding effect was recorded at the corner and lateral zones of the brick. The SEM images exhibited the penetration and the bridging effect achieved by the dense mycelium. The EDX revealed the high concentrations of carbon, oxygen, and calcium at all the analyzed points on the SEM images from all three samples. An inverted relationship between carbon and oxygen as well as sodium and potassium concentrations were also detected, implying the active metabolic interaction between the fungal hyphae and the seashell-based biocomposite. Finally, the results of the SEM-EDX analysis were applied to design favorable tessellation and staking methods for the V3-LBM from the seashell–mycelium composite to deliver enhanced biowelding effect along the Z axis and the XY axis with <1 mm tessellation and staking tolerance.

## 1. Introduction

### 1.1. Elastic Biodigital Brick: V3 Linear Brick Model

The geometrical design of the biodigital brick linear model V3 has been reported to achieve the best elasticity performance in comparison to the other biodigital and standard clay bricks [1,2]. It has achieved the second-best pre-cracking elasticity under (150 N) and the highest post-cracking elasticity under 200 N, while achieving the second-best compressive strength until 170 N. The current study exploits the formal efficiency of the V3 linear model in developing 3D-printed, seashell-based biocomposite material that is subject for biowelding with mycelium, where the brick’s bioreceptivity will contribute to its material coherence through bio-welding. This proposes the hosted bioactive agent as a binder agent that performs biowelding by its growth. 

The aim, thus, was to study the mutual effect of the formal design and bioactive material properties in achieving multi-scale sustainability including controlled material deposition, eco-friendly biomanufacturing and 3D printing, bioreceptivity, food waste recycling, and mycelium biowelding efficiency. This goes beyond literature to test the capacity of mycelium to bind non-agro–based substrates and to penetrate and bind lattice linear forms; not surface-based topologies. This interaction between the mycelium as the bioactive agent and the hosting material–geometry context is intended for testing multiple parameters, including developing a printable biocomposite material from food waste with a high shape fidelity and a lower degradability rate for applications in the built environment; testing the biocompatibility of the biocomposite material with the hosted bioagent; testing the bioreceptive capacity of the V3 elastic biodigital brick geometry for mycelium growth and penetration; and testing the chemical–physical reactions between the fungal culture and the biocomposite host and its effect on the overall morphology and shape fidelity of the brick itself.

In the following section, an overview is presented on the state of the art of utilizing biocomposite materials from different sources and types as sustainable building materials, as well as the mycelium biowelding effect in mycelium-based biocomposite materials. This is followed by the experimental study results and discussion of the developed seashell–mycelium biowelded brick through the SEM and EDX chemical analysis of the developed material reactions and coherency. Finally, a report of the materials and methods is conducted in the current study.

### 1.2. Biocomposite Materials: Fiber Based to Platelet Based

Recently, there has been an increasing interest in biocomposites within the construction industry thanks to their multiple benefits. For example, they are developed from renewable resources or recycled materials, in addition to their affordability, biodegradability, and competent physical properties such as light wight. A biocomposite is composed from a matrix and a reinforcement which is usually made of natural fibers. Natural fiber biocomposites have emerged as an environmentally friendly and cost-effective alternative to synthetic fibers. They are advantageous for their low density, resulting in a higher specific tensile strength and stiffness, as well as their lower manufacturing costs and easy production. This enables the wide range of applications of the natural fiber-based biocomposites, especially in the construction industry for use in architectural structures. Furthermore, their hollow structure enables their application in thermal and acoustical insulation, while the matrix is formed by polymers derived from renewable and non-renewable resources. It provides protection for the reinforcement fibers from environmental degradation and mechanical damage as well as a binder agent to adhere the fibers together and to transfer the loads on it. Biofibers originally are derived from biological origins, such as hemp and sisal [3]. This typically classifies biocomposites as being non-wood or wood-based fibers. The non-wood or natural fibers are better in their physical and mechanical properties, as well as their length and high cellulose content that deliver a high tensile strength; however, they are hydrophilic due to their content of hydroxyl groups (OH), which cause swelling and voids at the interface of the composite that affect their mechanical properties and dimensional stability. This is unlike wood fibers, which have a lower degree of cellulose crystallinity with varied flexibility and strength between softwood fibers (long and flexible) and hardwood fibers (shorter and stiffer). However, these wood fibers could be recycled or non-recycled, which hinders their sustainability. Wood-based fibers with more than 60% of their content being wood hinders the conservation of wood-trees and forests, in addition to the complicated processing of wood-based fibers.

These biocomposites suffer from lack of compatibility between synthetic resin and natural fibers [4,5,6]. This deters from the overall coherence and mechanical properties of the biocomposite, demanding a new method for developing biocomposites with enhanced coherency by creating a homogenous paste so that the reinforcement is well mixed with the matrix on a micro scale. This leads to a shift from fiber-based biocomposites to particle or platelet-based biocomposites. Seashell-based biocomposites are one possible solution, since seashells exhibit the organization of the martial microstructure in laminated platelets or particles on a micro scale [7,8]. This opens wider possibilities for integrating non-agro wastes in biocomposite material composition. Thus, in the current study, a seashell-based biocomposite material is developed from food waste and tested for biowelding by *Pleurotus ostreatus* mycelium. The material composition, printability, rheology, biocompatibility, and the physiochemical reactions with the mycelium were tested, as well as the mutual effect of the geometry and the material on the biowelding process by the mycelium and vice-versa. Further mechanical tests of this seashell–mycelium-based biocomposite will be exhibited in a following study on the mechanical properties of the developed biocomposite (seashell–mycelium brick) from a material–geometry interaction point of view.

#### Seashell Biocomposite from Recycled Food Waste

Seashells were a popular ingredient in building materials in vernacular architecture. *Tabby concrete* is one example that was made from mixing burnt oyster shells to create lime, with water, sand, ash, and broken oyster shells. Originating from the North African Islamic Architecture building materials and transferred to the Iberic Peninsula in the Middle Ages, it was made into bricks or used as “oyster shell mortar” or “burnt shell mortar”. Numerous surviving examples of Tabby concrete can be found in vernacular buildings in Morocco and Spain with mild differences such as adding Spanish moss to the Spanish Tabby [9]. Interestingly, Tabby concrete is a man-made mimicry of coquina, the naturally formed sedimentary rock derived from shells and used for building. Later in the early 19th century, the Tabby concrete became popular in building construction, coated by plaster or stucco for protection.

Despite being a popular traditional building material, it was replaced by clay bricks in the construction industry, since clay bricks became more affordable were produced on a mass scale. Recently, the sustainability challenge in the construction industry triggers reducing the use of non-renewable materials with high carbon emission production processes and introducing more durable and eco-friendly produced materials. Thus, seashell biocomposites from recycled food waste of seashells and molluscs, which are rich in calcium carbonate, offer an eco-friendly and structurally sound alternative building material. Recently, limited attempts to develop and study seashell biocomposites focused on their mechanical and chemical properties as regenerative materials for regenerative medicine applications, replacing the expensive, invasive, and infectious practices such as grafts. For example, Razali, et al., developed PLA biocomposites filled with calcined seashell particles prepared by melt mixing technique using a twin-screw extruder to produce a homogenous dispersion of fillers within the biopolymeric matrix, with enhanced tensile modulus and strength compared to the typical PLA [10](Razali et al., 2021). Other studies proposed seashell composites as an alternative building material. For example, the mechanical and chemical properties of the mollusc shells were studied and evaluated for their calcium carbonate content, and so a proposition of the implementation of seashell-reinforced composites in the construction industry as an alternative for non-renewable materials [11] was made.

Another study tested the mechanical properties of developed seashell biocomposites from sintered seashell filler sourced from Mediterranean coasts’ seashells using heat-treatment with calcium oxide (CaO). This revealed the relation between the developed seashells’ filler wing-like morphology with its mechanical properties’ improvement and proved the effect of the seashells’ filler on surface microhardness of the biopolymers as well. The developed biocomposite contained 20 wt% of the seashell fillers to poly butylene succinate, poly lactic acid, chitin, chitosan, and poly(ɛ-caprolactone), to improve the biocomposites’ hardness values by 29.8%, 57.9%, 51.1%, 26.2%, and 73.4%, respectively [12]. Another study developed a 3D printing binder jet mechanism to bond seashell powder-based ceramic composites [13], varying the composition of the composite powders from 5% to 50% of the seashell powder to plaster and achieving the necessary green strengths within the binder-jet process conditions. This reached the optimum levels of seashell powder of 15–20% by weight in terms of the best compression strengths.

Another study proved the durability of partially replaced seashell cement prepared by grinding and burning bivalve clam seashells to produce seashell ash powder. This partially replaced cement by 5, 10, 15, and 20% by weight; these were tested and compared with a SCO that had 0% seashell ash powder (SCO) [14]. The use of seashell-based concrete mixes as a replacement of fine aggregates, coarse aggregate, or as supplementary cementitious materials [15,16] is recommended.

Many experimental studies have proposed the use of waste materials in cement manufacturing [17], especially seashell waste accumulating on coastal areas in seashell-concrete as cement replacement in bricks production [18]. The seashell waste improves the mechanical and physical properties of concrete thanks to its high calcium content [19]. However, in all these previously mentioned studies, seashells were used as a supportive or partial mechanical enhancer, but not the main reinforcement in a composite.

The significance of seashell waste as a potent sustainable construction material arises from its abundance, since every year, about 45,000 tons of waste seashells are produced around the world [14]. This waste develops smelly odors due to the degradation of the left-over flesh or the decomposition of the salts contained in the shells into gases such as H_2_S, NH_3_, and organic compounds such as amines [14]. The seashell waste sources are varied in their types, including oyster shells, mussel shells, scallop shells, periwinkle shells, and cockle shells.

The chemical composition of seashells varies according to the type of shells, source, and mineral composition of the water bodies. Despite the slight differences in their chemical composition, raw seashells are highly rich in calcium carbonate (CaCO_3_—95–97%) with small quantities of minerals and organic materials [20,21,22]. The seashell powder produced by burning the shells at high temperatures is rich in calcium oxide (CaO 52 to 57%) depending on the type and the composition of CaCO_3_ content of the raw shells [14], and the adopted burning method at 1000 °C [23] to convert it from CaCO_3_ to CaO. Further parameters include the shape of the particles and their textures, and surface area.

However, the bioreceptivity of cementitious materials was tested infrequently in literature. For example, a study on the bioreceptivity of cementitious materials tested the suitability of magnesium phosphate cement (MPC) materials to allow a rapid natural colonization compared to carbonated OPC samples. This was conducted by modifying the aggregate size, the w/c ratio, and the amount of cement paste of mortars made of both binders [24], proving the effect of pH, porosity, and roughness in achieving better bioreceptivity of green concrete walls [24]. Another study focused on modifying concrete structures to facilitate bioreceptivity and biodiversity by substituting cement binder and aggregates in varying proportions and combinations to enhance the primary bioreceptivity of concrete, either chemically or via micro topographical texture. This tended towards enhancing surface roughness for enhanced bioreceptivity, rather than the chemical tunning of concrete that is likely to be spatio-temporally limited for months [25]. Thus, there is a shortage in studying the bioreceptivity of seashell biocomposites, which indicates the need for proposing seashell-based bioreceptive material as a sustainable and environmentally friendly alternative construction material.

### 1.3. Mycelium Biocomposites and Biowelding

Mycelium biocomposites are widely applied to different areas, including construction and biomedical applications [26,27,28], due to their cost-effective and wasteless production methods. Mycelium can be grown on agro-waste substrates, composing a dense hyphal network of fine white filaments of 1–30 μm in diameter [29], integrating the agro wastes from pieces to continuous composites. These wastes are composed from cellulose, tannin, and lignin, along with other various proteins, lipids, and carbohydrates [30]. This produces the final product of the mycelium biocomposite after drying the mix at a high temperature for several hours to stop the growth of mycelium. Growth and processing parameters such as humidity and temperature rule the mycelium composite properties and consequent various applications. These mycelium–agro-waste composites enjoy enhanced mechanical properties for structural applications [31,32,33,34,35]. For example, mycelium-based foam and sandwich composites have been proposed as construction materials, as well as their use as synthetic planar materials like sheets and semi-structural materials for paneling, flooring, and furniture. Thus, the application scope of mycelium composites can be tailored by the customized growth of the fungal species, including the growth media and conditions as well as the further processing methods to achieve specific mechanical characteristics for specific applications either as structural materials or for acoustic and thermal insulation and fire resistance. For example, substrates, including rice husks and glass fines, significantly increase the fire resistance of the mycelium biocomposite [36,37]. In acoustic insulation, mycelium composites achieved over 70–75% acoustic absorption at 1000 Hz, where different mycelium biocomposite panels were tested using various substrates. The highest acoustic absorption was achieved by using 50% switchgrass and 50% sorghum. This lead to the development of composite acoustic panels that are cost-effective and biodegradable [32]. Furthermore, mycelium is also a source of different types of chitins and chitosan known for their competent mechanical properties [28].

Mycelium composites are ecofriendly in terms of their production methods as an alternative to the traditional construction materials [38]. On the other hand, the agro waste from the rapidly increasing annual consumption of agricultural products—which are usually discarded or burned—generate carbon dioxide and other greenhouse gases [38]. However, these agro-waste materials have been used in limited application as low-durability building materials, such as components in bricks and green concrete for low-rise buildings, insulation materials, or for non-structural applications like fillers for road construction [39]. A more sustainable approach to integrate these agro-waste materials in durable and efficient construction materials is therefore needed.

This integration of agro wastes as substrate for mycelium biocomposites is starting to gain interest, as exhibited in some recently published studies. For example, a recent study tested two different mycelium composites for construction applications. These were mycelium-based foam (MBF) and mycelium-based sandwich composites (MBSC) [40]. The MBF made by growing mycelium homogenously in agricultural wastes in small pieces [41] produced fibers that bind these pieces together to form a porous material [42,43]. The MBSC panels utilized natural fibers as the top and bottom layers, with the central core made of the agricultural wastes combined with mycelium to form a sandwich structure of higher bending rigidity [43]. Both MBF and MBSC have proven mechanical strength, lightweight, and environmental advantages in building insulation, [31,40,43,44]. As well as foams, mycelium glues the core material to the fibers though the interface generated during the mycelium growth to resist delamination at the material interface under shear force, leading to a strong composite board with high bending stiffness [43,45].

The mycelium mechanical properties are controlled by its branched hyphal filaments and the topology of the network structures [34] which increase the contact area with the complex porous substrate. Every single mycelium fiber is composed of an array of cells separated by a septum and enclosed within the same cell wall. Tiny holes in the septum allow for the rapid flow of nutrients, water, and other small molecules from cell to cell along with the mycelium fiber. The cell wall protects the mycelium and provides mechanical strength, and is composed of a layer of chitin, a layer of glucans, and a layer of proteins on the cell membrane [33]. Chitin, which is a complex polysaccharide of N-acetylglucosamine, is located on the cell membrane and gives structural strength to the cell walls of fungal hyphae. It has α-chitin polymorph that is abundant in both crustacean and fungal chitin [46].

Thus, the mechanical properties of mycelium biocomposites are defined by the fungal strain, the substrate structure controlling the matrix mechanics within the composite, the growth conditions, the post-growth processing methods, and the water content of the final composite. Considering the competent mechanical properties and sustainable value of recycled seashells from food waste, the current study aims to develop a seashell–mycelium biocomposite material, combining the strength of the calcium-carbonate-rich seashells with the binding effect of the dense mycelium. The interaction between the seashell biocomposite and the grown mycelium to bind the separate 3D-printed profiles of the V3 linear brick model (V3-LBM) was tested and triggered an active biomineralization reaction of the biocomposite. The capacity of the mycelium in welding non-spatial lattice forms was also tested.

## 2. Results

### 2.1. Seashell Biocomposite Material Rheology

The seashell paste exhibited good printability, good shape retention post printing per layer, and layer-to-layer adhesion. However, it suffered from swelling in all directions with a 5 mm increase in the three dimensions of the 3D-printed brick that was lost post-drying.

The required scale-down of the V3 linear brick model design fit the printing bed dimensions of the paste extruder Ender 3D printer. Figure 1 exhibits the uniform scaling of the V3-LBM using the Rhinoceros Grasshopper to its customized dimension of 221 × 109 × 34 mm, which is 75% smaller than the original design model. Figure 1d–h exhibit the overall shape fidelity of the 3D-printed V3-LBM (post-printing and before drying) showing shape retention per layer as well as layer-to-layer adhesion.

As exhibited in Figure 1, the 3D-printed V3-LBM profile weighed 149.29 g post-drying; this profile was composed of five layers printed with the developed seashell biocomposite. The overall weight was calculated of a full V3 linear brick model that was originally composed of 34 layers according to the layer height of 1 mm; considering the scaled width of the brick, which is 34 mm, the total weight should be 1015.172 g of this scaled 3D-printed V3-LBM from the developed seashells composite. This is less than the weight of the same scaled V3-LBM model printed in clay. Taking into consideration that a 3D-printed clay V3-LBM that is 75% smaller than the original design should weigh 1658.66, this indicates that the seashell biocomposite V3-LBM is 40% lighter than the V3-LBM clay brick of the same size. Furthermore, the seashell percentage in each profile after the evaporation of the water was 80%, estimated by dividing the original weight of used seashell in the recipe by the weight of each profile post-drying: 120 g/149.29 = 0.8.

The dry seashell biocomposite exhibited large particle sizes ranging between 100–500 µm as exhibited in Figure 2. This demonstrates the material micro texture comparison between the clay brick and the seashell biocomposite brick of the V3-LBM, showing the textured seashell biocomposite brick with big platelets or particles.

### 2.2. Biowelding by Mycelium (SEM and EDX-Chemical Analysis)

After the first week of cultivating *P. ostreatus* between the three separate 3D printed profiles of the seashell biocomposite V3-LBM, the fungal hyphae started to form dense networks emerging from each piece of the seeded pieces of the *P. ostreatus*. Figure 3a–c exhibit the seeding process of *P. ostreatus* between every two 3D printed profiles. Figure 3d exhibits the culture development after one week where the fungal hyphae started to develop dense mycelium. The overall brick was swollen by the effect of the residues of cellulose-based media. After four weeks of cultivation the brick starts to get stiffer and dryer with denser mycelium between the three profiles. As the brick regained 3 mm to 5 mm increase in all 3 dimensions post inoculation with *P. ostreatus*. however, the final evaluation of the dimensional change of the seashell–mycelium biocomposite post drying revealed maintaining the same dimensions of the dry seashell-biocomposite before inoculation with ±1 mm dimensional change at each dimension. Figure 3e exhibits the sampling zones that were investigated by SEM-EDX analysis, exhibited in the following figures.

The sampling method was designed to detect the biowelding effect by the mycelium when needed the most, such as the corners, the lateral zones and in the middle zones of the biowelded brick, since the brick model is following linear, irregular lattice form. Demonstrating the crucial role of the biowelding effect by the mycelium in the current study and exceeding previous literature of surface-based biowelding effect that is an easier homogenous spatial context for the fungal hyphae to extend in all directions around 360 degrees. Unlike the current study, where the challenge is to test the mycelium customized biowelding effect. Exploiting the natural intelligence of fungal hyphae in searching nutrients and clustering to bioweld the separate three profiles of the V3-LBM with its lattice linear form. Furthermore, to detect the capacity and decision of the fungal hyphae either to penetrate the seashell-based biocomposite, or to extend over the vacant zones between the curves of the V3-LBM. Reflecting the tight design of the experimental study to investigate novel capacities of the fungal mycelium for customized biowelding of lattice structures.

The results of the scanning electron microscopy revealed that the best biowelding effect was found where most needed: on the corners of the brick as exhibited in Figure 4, which exhibited the SEM images of the S (side) sample from different orientations. This was followed by the lateral zones of the L sample from different orientations as exhibited in Figure 6, while the biowelding effect was less in the middle sample M as exhibited in Figure 5 due to the minimum surface extension of the curvature angle that the sample was taken from.

Figure 4 exhibits the SEM images of the side sample which is located on the left side top corner of the V3-LBM. The investigated areas of this sample detected the density of the fungal hyphae (mycelium) between every two layers of every two profiles: between middle and top, and middle and bottom profiles. These areas were examined from different orientations as labelled in the different SEM images in Figure 4 as follows: S-mb is between the middle and bottom profiles, S-mt is between the middle and the top profiles, S-S is the side surface of the welded profiles, S-h is the hyphal networks on the surface of the welded profiles.

As exhibited in Figure 4 from S-mb to S-mb_4_, the high density of the fungal hyphae is proved on different levels of magnification from 75×, 100×, and 500×, where S-mb and S-mb_1_ clearly detect the fungal hyphae bridging the two layers of the two separate profiles of the middle and the bottom. S-mb_2_, S-mb_3_, and S-mb_4_ show a closeup of these fungal hyphae bridges exhibiting the dense hyphal network and its strong anchorage and diffusion on multiple spatial points along the topologies of both layers of the bottom and middle profiles.

Similarly, the S-mt_1_ to S-mt_4_ SEM images of the biowelded middle and top profiles exhibit the bridging and welding effect of the fungal hyphae. S-mt_1_ and S-mt_2_ exhibit a 100× overall image of the biowelded middle and top profile showing the fungal hyphal bridges at multiple points on the topologies of both layers from the two profiles, while S-mt_3_ and S-mt_4_ exhibit the dense mycelium in these bridges under 500× magnification. S-s_1_ to S-s_3_ exhibit the surfaces of the biowelded layers between the middle and top profiles under 500×, 2000×, and 2857×. S-s_2_ under 2000× exhibits the penetration of the surface of the seashell-based biocomposite by the fungal hyphae, showing the active growth and colonization by the fungal spores that are spread (embedded on the layer’s topologies) as well as the fungal hyphae that cut through the surface of the material with a full penetration effect. The density of this penetrating mycelium is exhibited in a S-s_1_ SEM image under 500×, while S-s_3_, under 2857×, exhibits a closeup of part of the hyphae–biocomposite bridge showing the complete diffusion and the active colonization of the fungal spores on the bridge.

Finally, S-h_1_ to S-h_3_ focus on the dense hyphal network that composes the mycelium penetrating the seashell-biocomposite layers responsible for the biowelding effect between the middle and top profiles of the V3-LBM 3D printed profiles. S-h_1_ under 500× exhibits the overall dense mycelium, while S-h_2_ and S-h_3_ exhibit the fungal spores and hyphae under 2000× and 5000×, respectively.

The biowelding effect by the mycelium in the middle zone of the biowelded seashell–mycelium brick is exhibited in Figure 5. of the sample M taken from the middle interior zone of the brick as exhibited in Figure 3. The SEM images of the M sample labelled as M-s, M-s_1_ and M-s_2_ exhibit the biocomposite top profile (bottom layer) from its upper side surface that is biowelded with the surface of the middle profile (top layer), under 100× and, 500×, respectively. As exhibited in these images, the mycelium penetration of the biocomposite profile layers and the biowelding effect were low due to the heterogenous surface extension and the curvature angle of the curves of this part of the V3-LBM. On the other hand, this part of the curve is adjacent to the other neighboring curves (Figure 3) that could have facilitated the fungal hyphae ariel extension overall of the brick’s topology despite the voids between the curves. This did not happen; on the contrary, the fungal hyphae extended precisely on the curves of the V3-LBM, avoiding the void areas of the V3-LBM topology. This intelligent behavior of the fungal hyphae solving the maze of the lattice curvilinear geometry of the V3-LBM proves the fungal hyphae preference to penetrate the solid surface support as spatial extension even in minimum distance tolerance between the neighboring curves (2 mm) of the V3-LBM interior middle zone of the sample M (Figure 3). This proves that biowelding by mycelium can be adopted to weld curvilinear lattice forms, not only surface and volumetric forms, and can maintain good resolution of the design features. When moving down to the bottom surface of the same bottom layer of the top profile biowelded with the top layer of the middle profile, the density of the mycelium increased significantly with multi hyphal scales as exhibited in M-s_3,_ and M-s_4_ under 1000× each. This describes a favorable customized biowelding effect where the fungal hyphae densify precisely when needed most to perform the biowelding on the Z axis along the separate profiles and their layers. This is where the generated energy from the fungal metabolic activity is exploited in the fungal hyphae extension and penetration of the seashell biocomposite solid surfaces and binds them on the precise contact locations. This avoids wasting energy on building fungal hyphae over void areas between the linear curves of the V3-LBM. Furthermore, moderating the density of the mycelium at the conjunction points allows for a more customized energy-controlled extension and penetration. This proves that fungal hyphae favor solid surface penetration as a support for the fungal culture rather than a void (arial) non-supported extension. Finally, M-s_5_ exhibits the upper layer surface of the bottom profile biowelded with the bottom layer surface of the middle profile, under 1000×, exhibiting a moderate to high density mycelium biowelding the middle and bottom profiles.

Finally, the second-best biowelding performance was detected through the sample L of the lateral zone of the biowelded-biocomposite V3-LBM (Figure 3). This is manifested by welding the three separate 3D printed profiles of the V3-LBM as follows: between the top and middle profiles as exhibited in Figure 6 in L, L-t, L-t_1_, L-t_2_ and L-m under 65×, 75×, 500×, 2500× and 500×, respectively. In Figure 6-L and L-t exhibit the dense mycelium bridging the two layers, the bottom layer from the top profile and the top layer from the middle profile, while maintaining the coherence of the seashell biocomposite. L-t_1_ under 500× and L-t_2_ under 2500× exhibit the intertwining reciprocal dense mycelium forming the bridges biowelding the bottom layer of the top profile and the top layer of the middle profile. Similarly, L-m under 500× exhibits the dense fungal hyphae bridging between the top layer of the middle profile and the bottom layer of the top profile of the 3D-printed biocomposite V3-LBM. This shows the extension, emergence, and penetration of the fungal hyphae to the side surfaces of these two separate layers, while L-m_1_, and L-m_2_, under 2000×, exhibit the dense fungal hyphae penetration to the biocomposite mass of the top layer of the middle profile and extension to the bottom layer of the tope profile. L-m_3_ and L-m_4_ exhibit details from L-m_1_ and L-m_2_, respectively, under 5000×, exhibiting the active colonization of the fungal spores and dense hyphal net. Similarly, L-b, L-b_1_, and L-b_2_ under 500× and 1000×, respectively, represent the dense fungal hyphae of *P. ostreatus* between the bottom layer of the middle profile and the top layer of the bottom profile. This exhibits the dense integrated mycelium with the seashell biocomposite. This biowelding effect is denser than the other zones of the same sample outweighing the biowelding efficiency between the top and middle profile. This indicates that the fungal hyphae penetration effect is congruent with the gravity. L-b_3_ under 1000× exhibits the integration and penetration of the fungal hyphae to the biocomposite mass, achieving a cohesive binding effect between the middle and bottom profile. Finally, L-b_4_ under 1000× and L-b_5_ under 5000× exhibit the fungal hyphae and spores of the dense fungal culture on the top layer of the bottom profile that is binding and biowelding the biocomposite layers and profiles.

The EDX chemical analysis of the three samples (S, M, and L) detected the chemical composition comparison between the seashell biocomposite and the biowelded layers that encompass the mycelium as the binding agent between the separate 3D printed profiles of the V3-LBM. Thus, Figure 7 exhibits selected SEM images from the three different samples (S, M, and L) that were chosen to offer an overall insight into the varied chemical composition between the seashell biocomposite, the mycelium, and the biowelded layers encompassing the penetration and integration of the fungal hyphae within the biocomposite mass. Each SEM image includes multiple points on the examined topology to analyze on a micrometer scale the distribution and variance of the chemical elements encompassed in the topology of each zone. This excludes gold (Au) from the SEM sample preparation, and hydrogen (H), which cannot be represented in the SEM-EDX since they are uncharged into the metal inside the instrument due to the SEM vacuum operation. The chemical elements present in each zone at different points are exhibited in a table below each SEM image, representing the different concentrations of the various elements at each point. Figure 7 S-mb_3_ exhibiting the fungal hyphae bridging the two layers of the two separate profiles of the middle and the bottom, have been chemically analyzed at five different points along the topology between the bottom layer of the middle profile and the top layer of the bottom profile. Point 1 on Figure S-mb_3_ is located on the seashell-based biocomposite mass, while points 2 and 3 are located on the bridging mycelium that is integrated on top and within the seashell biocomposite mass. Points 4 and 5 are located on the fungal hyphae. As exhibited in the table of S-mb_3_, **carbon** atoms concentrations in the five different points were similar in ratios ranging from 52.78 to 67.89% with the highest concentration at the 4th followed by the 5th point on the S-mb_3_ topology, which mainly represents the high density of the fungal hyphae with their chitin rich cell walls. Similarly, **oxygen** atoms concentration was recorded at all the different 5 points with the maximum concentration of 41.32% at the 2nd point followed by 32.22% at the 3rd point. This is justified by the location of these two points on the conjunction between the biocomposite mass and the fungal hyphae, where the presence of oxygen atoms is interpreted by the presence of calcium carbonate from the seashell biocomposite, the cellulose media and the fungal hyphae chitin. Similarly, carbon atoms concentrations were present at all the four points on the S-s_2_ topology that represents the fungal spores’ colonization of the biocomposite mass, with the highest carbon concentration of 66.40 at the 2nd point and close range of similar concentrations at the S-mb_3_ topology. This pattern of carbon concentrations repeats in S-s_3_ that exhibits the integration of the fungal hyphae and the biocomposite surface, recording the highest concentration of 72.13% at the 3rd point on S-s_3_. Similar concentrations of carbon exist on L-t_1_ with the maximum of 66.90% at the 4th point. Comparing the different concentrations of carbon between (S-mb_3_, S-s_2_, S-s_3_, and L-t_1_), it is found that the fluctuations of carbon concentrations at the different points at each image reach the highest values congruent with the presence and location of the dense fungal hyphae as follows: (S-mb_3_: 4th point; S-s_2_: 2nd point; S-s_3_: 3rd point; L-t_1_: 4th point). This implies that the presence of carbon in this case is increased by the active metabolic activity of the fungal cells consuming the substrate and producing enzymes to build their hyphae cell walls adding to the strength of the seashell biocomposite. This is furtherly justified by the lower concentrations of carbon in M-s which mainly represents the biocomposite mass with low mycelium density recording the highest concentration of 47.76% at the 1st point. As well as the fluctuating carbon concentrations exhibited in L-b with the maximum carbon concentration existing at the 2nd point which is located on the fungal hyphae zone recording 64.26%.

Oxygen concentration had an irregular and inverted relation with the carbon concentration at each point, as exhibited in Figure 8a, where higher carbon concentrations correspond to lower oxygen concentrations and vice versa. For example, in S-mb_3_, the lowest concentration of carbon of 36.62 at the 2nd point corresponds to the highest concentration of oxygen of 41.32 at the same point, and the highest concentration of carbon of 67.89 at the 4th point corresponds to the lowest concentration of oxygen of 20.01 at the same point. This inverted relation can be detected also in S-s_3_, where the lowest concentration of carbon of 51.76 corresponds to the highest concentration of oxygen of 21.34 at the 1st point. This can also be detected in L-t_1,_ where the highest concentration of carbon atoms of 66.90 at the 4th point corresponds to the least concentration of oxygen of 22.65%, and the least concentration of carbon of 47.75% at the 1st point corresponds to the highest concentration of oxygen of 27.93% at the same point. While this pattern is exhibited with less regularity and more fluctuations in S-s_2_, M-s, and L-b. In M-s, which mainly represents the seashell biocomposite, carbon concentrations of 47.76%, 46.05%, 45.52%, and 38.63% correspond to oxygen concentrations of 29.30%, 48.89%, 40.56%, and 33.45%, respectively. This implies more relativity between the carbon and oxygen concentrations than all the other analyzed topologies. This proves the reaction between the fungal hyphae and the seashell biocomposite calcium carbonate, where the highest concentrations of carbon can be detected in the densest mycelium zones, accompanied with the least amount of oxygen that is more likely to be sourced from the calcium carbonate of the seashell biocomposite and the cellulose-based substrate. This implies that there is an active metabolic activity of the mycelium that breaks down the calcium carbonate and builds its dense mycelium. This also justifies the full penetrative yet binding effect by the mycelium to the seashells composite. On the other hand, the pattern of carbon-oxygen concentration is less likely referred to the presence of cellulose as a remnant from the growth media for two reasons; first, the growth media was distributed evenly everywhere between each two separate profiles, second, cellulose (C_6_H_10_O_5_)n contains almost equal amount of carbon and oxygen atoms, which implies that there would have been almost equal concentrations of carbon and oxygen. This implies that cellulose acted as a primarily catalyst substrate for boosting the growth of the fungal hyphae until it penetrates and digests the seashell biocomposite.

The concentration of **calcium (Ca)** recorded its peak at the 1st point followed by the 2nd point on the S-mb_3_ topology with 2.58% and 1.92%, respectively. This is congruent with the location of these two points mainly representing the seashell-based biocomposite which is rich in calcium carbonate CaCo_3_. However, the 3rd high concentration of the calcium atoms on S-mb_3_ was recorded at the 5th point with 1.27% concentration, which implies the diffusion of the mycelium through the biocomposite mass. This is supported by the high concentrations of Ca at all the four points on M-s that represents the seashell biocomposite with the highest concentration of 20.30% on the 2nd point, followed by 9.60% on the 1st point and 5.05% on the 4th point on M-s. These Ca concentrations on M-s are the highest comparing to all the other EDX analyzed SEM images, where they ranged from 0.91% on the 3rd point on S-mb_3_ to 4.23% on the 4th point on S-s_3_. Ca concentrations were regular in L-t_1_ that exhibits the fungal hyphae integration within the seashell biocomposite where Ca atoms were found at all the four examined points with higher concentrations than S-mb_3_. This was not the case for S-s_2_, S-s_3_ and L-b where Ca was found in some points only, with higher concentrations detected at points located on the biocomposite more than the fungal hyphae. As it is exhibited on the 4th point on S-s_2_ with 4.07% concentration, 4th point on S-s3 with 4.32%, and 4th point on L-b with 3.98%. Although this reflects that Ca presence is connected mainly to the seashell biocomposite, where even concentrations of Ca exist on all analyzed points on M-s and L-t1 and almost all points of S-mb_3_. However, the highest concentrations of Ca were found on M-s, S-s_3_, and S-s_2_. From the uneven concentrations of Ca on S-s_3_, S-s_2_ and L-b, it is implied that there is an active metabolic reaction where the fungal hyphae break down the seashell calcium carbonate. This is congruent with the carbon and oxygen concentrations pattern that reached to the same conclusion. This proves that mycelium can be cultivated on non-agro substrates such as seashells. This in turn opens wide possibilities for utilizing calcium carbonate as a bioreceptive building material as well as using mycelium as decomposer of food wastes from seashells.

**Nitrogen (N)** was recorded on the 2nd point on S-mb_3_ with 18.64% concentration, as well as the 1st point on S-s_2_ with 11.48% concentration, 1st point on S-s_3_ with 14.64% concentration, 1st point on L-t1 with 13.47% concentration, and on L-b on the 3rd and 5th points with the highest concentration of 15.69%. It was absent in M-s. It is noted that nitrogen concentration is always located on the points that are on or close to the fungal hyphae. This indicates the source of nitrogen is more likely to be the P. ostreatus mushroom pieces of the inoculum, as well as proves the favorable role of the detected nitrogen concentration in developing the fungal hyphae, and the role of the fungal hyphae in fixation of this element.

The equilibrium between **sodium (Na) and potassium (K)** recorded in the S-bm_3_ is interesting for further studying the fungal metabolic pathways in reaction with seashell biocomposite. Sodium (Na) was present only in the three first points that are located more in the seashell-based biocomposite zone, with the highest concentration of 3.19% at the first point, while potassium (K) was inversely present only in the other three points (3, 4, and 5) with the highest concentration of 2.03% at the 4^th^ point, which mainly represents the mycelium. The presence of sodium in the biocomposite zone can be justified by the seashells containing traces of sodium salts or sodium oxide. This is unlike the case of potassium that is less likely to be contained within the seashell biocomposite since it is completely absent in the 1^st^ and 2^nd^ points on S-mb_3_, which are mainly representing the biocomposite mass. Thus, the potassium (K) detected in the mycelium dense zones could be justified only in relevance to the fungal culture metabolic activity, fixated by the growth of the fungal culture. This aspect holds a further potential in analyzing the role of *P. ostreatus* in potassium fixation as well as the interaction between sodium and potassium in the fungal metabolic pathway. Since the pattern of sodium and potassium occupation and concentration at the different points on S-bm_3_ topology implies a sort of trade-off between the two elements. This is supported by the EDX analysis of S-s_2_, S-s_3_, and M-s, where on S-s_2_, potassium hardly exists with a 2.31% concentration on only the 4th point that represents the penetration of the fungal hyphae to the biocomposite mass. Sodium did not exist at any point on S-s_2_. On S-s_3_, potassium was present at three points with almost even distribution with 1.65% as the highest concentration at the 4th point, while sodium only existed at the 1st point on S-s_3_ with 1.19% concentration. Moreover, on M-s that is mainly representing the biocomposite mass, sodium existed at the 2nd, 3^rd^, and 4th points with almost even distribution and considerable concentrations between 2.09 to 2.40% compared to S-mb_3_ that encompassed the highest concentration of Na of 3.19% at the first point. Potassium was absent at the examined zone on M-s. Similarly, sodium was almost evenly distributed at all the tested points on L-t_1_, with concentrations from 1.35% at the 1st point to 1.68% at the 2nd point on L-t_1_, in the absence of potassium as well. L-b did not incorporate neither sodium nor potassium. As exhibited in Figure 8b, it is obvious the inverted relationship between sodium and potassium where sodium was found mainly related to the seashell biocomposite mass with higher concentrations and even distribution as in S-mb_3_, M-s, and L-t1, while being absent in S-s_2_ and L-b and only with a trivial concentration existing in one point on S-s_3_. Inversely, potassium existed almost at the points or zones that did not encompass any sodium as on the 4th and 5th points on S-mb_3_, and the 2nd and 4th points on S-s_3_ and was completely absent in M-s, L-t_1_ that involved an even distribution of sodium, and L-b. Thus, it can be concluded that sodium existence is based on the seashell biocomposite traces of sodium oxide or other sodium-containing compounds, while potassium is found relevant to the metabolic activity of the dense mycelium.

Finally, **Chlore Cl** was only trivially present in the 2nd point on M-s with 0.98% concentration which might be related to some salt traces in the seashell-based biocomposite.

Generally, the EDX analysis exhibits the dominant presence of carbon, oxygen, and calcium at all the points on all the analyzed SEM image. This implies the homogeneity and integration of the seashell–mycelium biocomposite. It is worth mentioning that the scale of the examined zones does not influence the result of the analysis, since the EDX was conducted on representative SEM images from different samples and zones and with varied scales from 20 µm to 1 mm as exhibited in the scale bars on each SEM image in Figure 7. Furthermore, comparing the wights of different elements with their atoms concentrations exhibits congruency which reflects the accuracy of the EDX analysis and its results.

### 2.3. Seashell–Mycelium Biowelded Brick-Tessellation & Active Biowelding

The capacity of mycelium to bioweld curvilinear forms is proven by solving the maze of the coiled curves of the V3-LBM, performing biowelding effect where needed. By penetrating and extending inside the seashells-biocomposite rather than extending over the void areas of the design. This customized intelligent biowelding activity of the mycelium can be exploited in active biowelding on a collective scale (e.g., a wall). And applied either as a construction method in replacement of traditional binders as mortar, or in a self-healing regenerative building material. Interestingly, mycelium in this case digests and induce the recompositing of calcium carbonate in an active oxidation-decomposition-biomineralization loop. Moreover, chitin in this case will source not only from the fungal hyphae but also from the seashells, which offers a self-healing strong material encompassing chitin and calcium carbonate through active mineralization. This aspect will be exhibited in a following study along with the mechanical tests of the developed seashell–mycelium biocomposite material. The mycelium intelligent biowelding based on solid surfaces penetration rather than void arial extension should be considered when designing the staking and tessellation of the V3-LBM from seashell–mycelium biocomposite. Focusing on maximizing the contact surface area between every two tessellated and staked bricks. On the other hand, the curvilinear lattice form of the V3-LBM that is varied in its topology and solid and void alterations offers only some specific contact points where the contact surface is considerable. Furthermore, the results of the SEM images of the different samples S, M, and L, where the maximum biowelding effect was recorded on the borders of the V3-LBM, are considered. Since the samples S and L were the best biowelded, congruent with the spatial analysis of the V3 linear form composed from the coiled curves. As exhibited in Figure 9. the most potent zones for staking the V3-LBM are located closer to the borders and corners highlighted in the circled zones.

The staking and tessellation organizations and orientations of V3-LBM from seashell–mycelium biocomposite is exhibited in Figure 9. Demonstrating the lateral zone of the brick where it is proven the high efficiency of the biowelding by mycelium, thanks also to the geometry of the V3 linear model at this zone, that offered short curves meeting on multiple contact points that supported the growth of the dense mycelium. This axis is the most potent for mirroring the tessellation of the bricks, to increase the contact points along the axis from both sides with <1 mm tolerance distance between every two bricks either tessellated or staked. Thus, offering further support for the growth of mycelium and its biowelding effect to the seashell biocomposite. This is exhibited in Figure 9a–c. Furthermore, it is also recommended to stake the bricks on top of each other exactly without a shift to facilitate the penetration of the fungal hyphae in direct straight direction which will result in an increased compressive strength along the Z-axis (vertically), without hindering the mycelium biowelding effect by shifting the bricks. Since shifting the bricks on top of each other can result on obstruction of the growth of mycelium when it meets void areas in the bottom or top layers’ bricks, resulting in the loss of the continuous biowelding effect along the Z-axis.

## 3. Discussion

The search for new sustainable and high functional biocomposite materials with multi-scale sustainability in the construction industry has been under spotlight recently. Especially, in terms of low energy-consumption production methods. This motivated the current research objective to develop seashell–mycelium biocomposite using 3D printing-past extrusion and biowelding as an integrated biomanufacturing approach for domestic, decentralized, and affordable construction techniques [2]. For a 3D-printed-biowelded, seashell–mycelium brick, as an affordable and available construction technology.

The biowelding capacity of mycelium was tested to penetrate and bind seashell biocomposite into a biowelded seashell–mycelium brick. The novelty of this study is employing mycelium biowelding the lattice and curvilinear form of the V3-LBM. Exploiting the intelligence of the fungal hyphae in biowelding the separate 3D printed profiles of the V3-LBM with their limited contact surface area along the curves of its lattice form. Beyond most of the previous literature which focused on surface-based and volumetric designs for mycelium biocomposite applications [28,47,48], mycelium-bound porous structures applications [49,50] where the bound structures are more of volumetric type. This exceeds reinforcing mycelium bound composites with lattice wood structures [51], or the direct extrusion 3D printing of mycelium bound biocomposites in free form structures with more volumetric scale than linear [52,53]; or even testing the mycelium biowelding effect on lattice bricks with more regular forms and wider thickness of the profiles of 5 mm to 9 mm [54], which is five times more than the nozzle size of 1.8 mm used in the current study. The current study used mycelium as the main binding agent, without temporal or primary supporting structures and in a fine scale of lattice curvilinear form that is <2 mm layer width, 1 mm layer thickness, and with equal solid-void distribution in the V3-LBM topology. Results of SEM analysis revealed the customized biowelding effect as that the fungal hyphae solved the maze of the V3-LBM coiled curves, achieving biowelding precisely where desired between the three separate 3D printed profiles of the brick.

The second point of novelty in the current study is testing the capacity of mycelium to penetrate, and weld seashell-based biocomposite which was not reported in previous literature. Detecting the reaction between the fungal hyphae and the seashells composite. This was proven by the SEM-EDX analysis indicating the reaction between the fungal hyphae and the seashell’s calcium carbonate while maintaining cohesion to produce a seashell–mycelium brick. Beating previous attempts that tested mycelium biowelding of 3D printed lignocellulosic biomass, post-extrusion [55].

Mycelium biocomposites have witnessed significant integration in the construction sector recently. Offering possible load-bearing materials applied in mycelium bricks and concrete with an average compressive strength of 5.7 MPa and 22.5 MPa, respectively [56]. Examples of mycelium bricks are abundant in literature [44,57], or as biopolymers for architectural cladding [58,59,60]. These projects have triggered the interest in understanding the capacity of mycelium to produce lightweight and mechanically competent building material. Numerous literatures focused on studying the various culturing methods, conditions, substrates, post-growth processing and their effect on the mechanical properties of the developed mycelium or mycelium bound biocomposites [61,62,63].

Since the mechanical properties of mycelium biocomposites control their applications. The network structure of mycelium within the composite is determined by the species of the fungi and the growth-substrate. This was manifested by comparing the varied mechanical properties of mycelium composites across various studies. The substrate affects the density of the mycelium-based composite, where the higher proportion of grain contained in the substrate will lead to a higher density and the higher material density of mycelium is mandatory to deliver high Young’s modulus and strength [64,65,66]. It was proved that specific substrate can improve the compressive strength and the flexural strength of the mycelium biocomposite that can be used as construction material [67]. However, most of these previous studies, usually used substrates from agricultural crop waste such as cotton, corn, wheat, and hemp, and flax residues as lignocellulosic wastes [40,43,68], varying their types, chemical composition and other growth media and conditions alterations. Although it is sustainable to recycle these agro-waste and employ them as a substrate for the growth of mycelium to produce homogenous biocomposite. However, main challenges as the hydrophilicity of the produced biocomposite and its decreased density make these mycelium biocomposites prone for poor water-resistance and swelling [69,70]. Shortening their life cycle and hindering their durability making them incompetent for long-lasting durable building material applications without the need for coating with other non-sustainable finishing materials. Additionally, the recent climatic change and world geopolitical events affected the global agricultural production and consequent agro-wastes due to the shortage in fertilizers and drought detected in many zones since 2022 [71,72]. Thus, prompting the need for an alternative substrate that is not hydrophilic, available, and sustainable being sourced from recycling food wastes.

Therefore, the current study proposes a novel methodology to combine seashell-waste, agro-wastes, and mycelium to produce seashell–mycelium biocomposite bricks. This exploits the mechanical strength of both mycelium’s chitin and seashells’ chitin while utilizing cellulose as a catalyst substrate to boost the growth of the mycelium and enabling it to penetrate the seashell biocomposite.

In the current study, the seashell biocomposite was developed from recycled food waste. Since seashell waste are produced annually with large amounts internationally, for example China produces about 10 million tons of waste seashells annually, and The European Union produces 600,000 tons of seashell waste that are dumped in landfills resulting in visual pollution, unpleasant smell, and microbial contamination due to microbial decomposition of salts into undesired gases [73,74,75,76]. Therefore, numerous studies have proposed recycling seashells and integrating them as fillers and aggregates in building materials such as mortar, cement, and concrete [77,78]. This is thanks to their high content of calcium carbonate that makes 95% of their composition [79,80], and to their richness in chitin that comprises 20–30% of their composition and could reach 60% according to their type and species [81]. Chitin has excellent properties of non-toxicity, film-formation capacity, biodegradability, and biocompatibility. It occupies the second place after cellulose in being the most abundant natural biopolymer that is adaptable and environmentally friendly [81]. Chitin is usually related to the presence of calcium carbonate and lipids [82]. These properties have promoted the reintroduction of seashells waste as building materials in the construction industry, as seashell biocomposites in the form of tiles or bricks [7,83,84]. Thus, in the current study, seashell biocomposite was developed as a base material consisting of 77.7–80% seashells powder of its wet to dry weight, respectively, with 100 µm particle size as the main reinforcement of the biocomposite material. Beyond the previous literature that used seashell powder as a partial reinforcement or fillers with up to 20% ratio to the total biocomposite [12,14] and congruent with [13] that used up to 50% seashells powder to the overall composite.

Mycelium from *Pleurotus ostreatus* was chosen to be the binding agent of the seashell-biocomposite 3D printed separate profiles of the V3-LBM for the following reasons: First, *Pleurotus ostreatus* is the most used in mycelium-bound biocomposites development [85]. Second, besides its tremendous significance in producing essential chemicals including a variety of enzymes [33,86,87], it is sustainable as an available, non-pathogenic, and edible fungal strain. Boosting the possibility of developing edible bricks for future applications in architecture. The fungal species determine the mechanical properties of the mycelium bound biocomposite, based on its growth yield, thickness of mycelium fiber, hyphal microstructure, and hyphal surface topography [33,40,43,68,88]. Generally, there are three categories of mycelium hyphae. The generative hyphae which are undifferentiated, thin walled with occasionally thickened walls with frequent septa and clamp connections and can develop reproductive structures. Skeletal which are thicker, longer, and rarely branched with few septa and lack clamp connections. And the binding hyphae which are thick-walled, often solid, and branched [35,89]. Although *Pleurotus ostreatus* is categorized as monomitic species in terms of its mycelium network typology. Indicating that it comprises only generative hyphae. However, the SEM results from various examined zones of the biowelded brick exhibited a variance in the scale and density of the fungal hyphae exhibiting more thicken-walled long hyphal fibers that could be categorized as skeletal or binding hyphae more than generative hyphae. This could be justified by the favorable media composed of cellulose and abundant amount of calcium carbonate. However, further genetic identification of the fungal strain is recommended for future studies to understand possible strains variations concerning their hyphal network typology and the effect of media constituents on the adaptability of the hyphal topology. In the current study, genetic identification was not conducted, as the main scope focuses on testing and evaluating the possible penetration, interaction, and biowelding of the seashell biocomposite by mycelium in general—which is the point of novelty of the current study—regardless of the specific differences between fungal strains that would affect the final biocomposite mechanical properties, which will be exhibited in a following study in details.

The biocomposite composition, the fungal strain, the growth conditions including the primary growth substrate and pH as well as incubation conditions of temperature and humidity were analyzed relatively as exhibited in the following paragraphs.

Recent literature proved that calcium carbonate is favorable as a media constituent in boosting the growth yield of *Pleurotus ostreatus* [90,91]. This is congruent with the current study that used adequate concentration of calcium carbonate contained in the seashells as a booster for the growth of *P. ostreatus,* and as the main source of calcium ions. This is reflected in the calcium concentration in most of the tested points on the varied topologies of the three samples S, M and L, ranging from 0.91% atoms concentration to 20.30%. The highest Ca concentrations were mainly on M-s compared to all the other EDX analyzed SEM images. It ranged from 0.91% on the 3^rd^ point on S-mb_3_ to 4.23% on the 4^th^ point on S-s_3_. Ca concentrations were also found regularly at all the tested points on L-t_1_ that exhibits the fungal hyphae integration with the seashell biocomposite, with higher concentrations than S-mb_3_. And on S-s_2_, S-s_3_, and L-b, Ca was found in some points only, with higher concentrations detected at points located on the biocomposite more than the fungal hyphae. This indicates the low-to-moderate Ca concentrations on all the examined SEM images except M-s, which have promoted the dense mycelium growth on these zones. This is congruent with [92]. By comparing the EDX results of carbon and oxygen concentrations on the different tested points of the examined SEM images representing the varied topologies of the biowelded seashell–mycelium brick. The increase in carbon concentrations was detected mainly on the dense mycelium zones accompanied by low concentrations of oxygen at the same points. Comparing the highest carbon concentration of 72.13% at the 3rd point on S-s_3_ which is representing mainly the dense mycelium_,_ and the lowest carbon concentration of 38.63% on the 2nd point of M-s that mainly represents the seashells-biocomposite, it can be deducted that the calcium carbonate contained in the seashell contributed to the fungal hyphae development and growth. This is further emphasized by the inverted relationship between carbon and oxygen concentrations that indicate an active metabolic activity between the fungal hyphae and the seashell-biocomposite in an active biomineralization loop reaction.

Cellulose played a crucial role as the catalyst for the initial growth of *P. ostreatus* with a 50% ratio to the seashell biocomposite. Boosting the growth and density of the fungal mycelium. Congruent with many literatures reporting that cellulose is the medium of choice for the best growth yield of *P. ostreatus* [33]. As it improves the mechanical properties of the mycelium of *P. ostreatus* producing four different combinations of the mycelium with stiff fibers [33]. Congruent with the dense multiscale fungal hyphae produced by *P. ostreatus* in the current study. Sustainably speaking, cellulose is the most abundant natural polymer available in all hardwoods and crop wastes. Thus, the contribution of cellulose as the initial carbon source served in the initial growth of the fungal hyphae that oxidates cellulose to produce hydrogen ions through complex enzymatic reactions [93], to be integrated in the breaking down of calcium carbonate as proposed in the following equations:(1)$$Fungal\;cells+{\left(C_{6}H_{10}O_{5}\right)}_{n}=n(H^{+})$$
(2)$${CaCo}_{3}+2H^{+}={Ca}^{2+}+{Co}_{2}+H_{2}O$$

In a chain reaction of calcium carbonate degradation to produce carbon dioxide, calcium ions and water. Then calcium carbonate is recomposed by the dissolute CO_2_ in water, giving bicarbonate ions, forming carbonate ions to combine with Ca^2+^ to produce again CaCO_3_ once the solubility limit has been exceeded [94]. This will be exhibited in detail concerning the reaction dynamics and pH conditions in a following study.

The considerable concentration of xanthan gum of 25% to the seashells’ ratio of the biocomposite might have contributed as an additional carbon source for the fungal growth, congruent with [95] that proved that xanthan gum boosted the growth of various fungal strains. Similarly, glycerol might have contributed as a supporting carbon source, despite its trivial concentration of 0.028% to the seashells’ ratio in the biocomposite [96]. The 0.012% of acetic acid concentration was mandatory to adjust the level of the biocomposite to pH 7 to provide favorable conditions for the mycelium growth.

Nitrogen is essential for mycelium growth and various metabolic activities particularly protein and enzymes synthesis [97,98,99]. As well as a constituent of the cell wall of many fungi, composed of β (1–4)-linked unit of N-acetylglucosamine [100]. Ortega et al. (1992) [101], proposed that nitrogen increase in mushrooms is related to the amount present in the initial substrate added to the nitrogen amount present in the inoculum, indicating a possible fixation of nitrogen by *Pleurotus ostreatus*. Sturion and Oetterer (1995) [102] also observed a nitrogen increase in the residual substrate, which ranged from 4% to 37% in the cultivation of *Pleurotus* spp. suggesting that this genus fixes nitrogen. In the current study, nitrogen concentrations at the different tested points of the biowelded seashell–mycelium brick ranged from 11.48% on the 1st point on S-s_2_ to 18.64% on the 2nd point on S-mb_3_. It was proven that seashells are rich in nitrogen [103], with an average concentration of 0.67% to 0.69% by total dry weight (0.22–0.28 g N/shell), with possible variations according to the location, season of harnessing, or growth method. This is despite the fact of this relatively high concentration of nitrogen in seashells, and despite the fact that a high concentration of nitrogen from organic sources was found to inhibit the growth of *Pleurotus ostreatus* [97]. However, there are not sufficient literature reporting the effect of nitrogen from seashells on the growth of fungal mycelium in general and *Pleurotus ostreatus* in particular. Moreover, the concentrations of nitrogen in the current study revealed their relevance to the fungal hyphae locations more than the seashells-biocomposite mass. This indicates that the found nitrogen originated from the first fungal inoculum and have been incorporated in the fungal mycelium development.

Sodium was not reported to have significant effect on the growth of *Pleurotus ostreatus*. However, ref. [104] reported that more than 3% concentrations of NaCl inhibited the growth of *P. ostreatus*. Although the sodium source in the current study is the seashells, however, it is not known in which form it is present. Nevertheless, the sodium concentrations found at the different tested points did not exceed 3.19% which was the highest concentration among all the tested points. Furthermore, sodium was not present at all the tested zones, since it was present only in the seashell biocomposite zones justified by the seashells containing traces of sodium salts or sodium oxide.

Potassium had an inverted relationship with sodium as it was present mainly in the zones that did not incorporate any sodium atoms. However, potassium concentrations did not exceed the highest concentration of 2.03% at the 4th point on S-mb_3_, which mainly represents the mycelium; and on S-s_2_ with a 2.31% concentration on only the 4th point, which signifies the penetration of the fungal hyphae to the biocomposite mass. This is absent on M-s, L-t_1_, and L-b, and exists almost on the zones that did not encompass any sodium as on the 4th and 5th points on S-mb_3_, and the 2nd and 4th points on S-s_3_. Consequently, it is less likely that the potassium is contained within the seashell biocomposite since it was detected in the mycelium dense zones. Hence, it could be justified only in relevance to the fungal culture metabolic activity, as fixated by the growth of the fungal culture. Or from the starter culture mushroom pieces that were used to grow the mycelium between every two separate profiles. While the trad off between sodium and potassium requires further investigation to prove the role of *P. ostreatus* in potassium fixation.

In the current study, the developed biocomposite of pH 7 provided a favorable environment for the *Pleurotus ostreatus* to grow, since the optimum growth of *P. ostreatus* reported in literature was between 7–8.5 pH level [105,106]. Temperature and humidity are important factors in controlling mycelium growth as well. The current study used room temperature 25 °C for the incubation of the fungal culture. This is supported by [107]. Moreover, relatively high humidity is favorable for the growth of mycelium. This was conducted in the current study by sealing the seashell–mycelium brick with plastic wrapping paper, which is congruent with [43] that maintained high humidity environment with 98% relative humidity for mycelium growth using a semi-permeable polypropylene bag, offering a high humidity and sterile environment for mycelium growth.

The substrate and the fungal species determine the final mycelium water content [108]. Usually, cellulose-based substrates increase the moist content of mycelium up to >60% [61]. Furthermore, the proposed biomineralization loop reaction between the fungal hyphae and the seashell biocomposite maintains moderate water content by producing water in the calcium carbonate breaking reaction. The water content of the final mycelium biocomposites in turn, controls their mechanics. Indicating the need of water removal to inactivate the mycelium growth and provide high and reliable mechanical performance. Although there is not a recommended percentage of moisture residual in literature, however, the drying process should be conducted until the fungal growth is terminated [40]. In the current study, the extracted samples were dried at 130 °C for 90 min on two consecutive cycles with 12 h between each cycle to halt the growth of the mycelium before examining the samples with the SEM.

Furthermore, water content moderation techniques in the mycelium network affect its final mechanical properties. [68] For example, hot pressing is one famous technique to squeeze water and air out of the porous mycelium network, reduces the porosity of the material, and increases the material density, leading to higher Young’s modulus and strength [109,110,111]. As well as reducing thickness and increasing fiber connection between the walls of the fibers at points of overlap [40,112]. However, this technique was not used in the current study for two reasons: first, the current study aimed to detect the penetrative effect of the fungal hyphae to the seashell biocomposite by SEM microscopy, thus excluding any processing technique that would reorganize or affect the orientation and physical characteristics of the detected hyphae. Second, the seashell–mycelium brick is not based on fibers only, since the seashell biocomposite is a particle or platelet-based composite that its reaction with the fungal hyphae required topological investigation first, which is conducted in the current study.

Thus, Scanning Electron Microscopy was used to detect on a micrometer scale the penetration of fungal hyphae to the seashell biocomposite at different points, zones and topologies of the three samples S, M and L, congruent with [33].

The SEM analyzed zones exhibited the penetration of the seashell biocomposite by the dense fungal hyphae. Thanks to the chitin incorporated in fungal hyphae; in the innermost layer of the fungi cell wall that can provide reinforcement and strength. The specific organization of the secreted chitin inside or outside the fungal cells with its [β(1–4) linked N-acetyl-2-amino-2-deoxy-d-glucose]units [66], form weak bonds between its fibers increasing the strength of the entire structure [113]. Although chitin is also incorporated in the seashells, the fungal chitin is advantageous in terms of sustainable unlimited production [66,114,115], not limited to a season or location. Moreover, the extraction process of fungal chitin is more straightforward [114,115,116], not requiring processing with strong acid to remove calcium carbonate and other minerals as the case of seashell’s chitin [114,115]. Moreover, the fungi-derived chitin generates a natural nano-composite structure by branched β-glucan, which provide rigidity to the chitin and produce strong fiber networks when extracted [114,117].

Chitin’s high tensile strength, poly-crystalline polymeric acetylglucosamine, is responsible for the tensile strength of the mycelium with reinforcing capacity strength to its fibrous network [118]. Thus, the fungal chitin provided the required strength for penetrating the seashell biocomposite, along with the degradation enzymes. The degradation enzymes in the current study played a crucial role not only in degradation of cellulose as the primary substrate but also in digesting the seashell biocomposite [119]. These enzymes include Laccases, peroxidases, oxidases, cellulases, and different glycosidases [120,121]. The initial degradation of cellulose occurs by cooperating oxidative and hydrolytic enzymes of the fungal strain. The degradation occurs on both levels: intracellular and on the outer cell envelope layer. Where the extracellular hydrolytic enzymes degrade polysaccharide, and the oxidative enzymes degrade open phenyl rings [122,123].

Moreover, hydrophobins located on the outer surface of the filamentous fungi cell walls [124] facilitated the hyphal growth and its interaction with the host environment, which is the seashell–biocomposite in the current study. Hydrophobins facilitated the fungal hyphae attachment to the solid biocomposite support [125]. In addition, hydrophobins forms an amphipathic membrane where the hydrophobic side is exposed to the exterior and the hydrophilic surface is combined with the cell wall polysaccharides [124]. Finally, glucans integrated functional proteins and skeletal chitin to form fungal cell walls’ structural components [126]. All these factors have contributed to the dense mycelium effect in biowelding the separate 3D-printed profiles of the V3-LBM from seashell biocomposite acting as a binding agent.

Finally, the density of mycelium affects its mechanical properties, as it was proved that mycelium exhibit linear elasticity at low strains [34]. In the current study, the high density of the mycelium was exhibited in the SEM images, however the variance between the seashells’ particle-based biocomposite and the fibrous mycelium requires further investigation about their mechanical behavior, which will be presented in a following study. Along with dimension stability post-fungal growth that will be detected in growth-related fashion. Since it was proven in previous studies, that when the mycelium content increases, the linear dimensional change increases. However, in the current study, dimensional change of seashell–mycelium biocomposite brick occurred when the brick regained 3–5 mm increase in all 3 dimensions post inoculation with *P. ostreatus*. however, the final seashell–mycelium biocomposite brick post-drying maintained the same dimensions of the dry seashell-biocomposite before inoculation. This reveals the significant role of the seashells-biocomposite to maintain dimensional stability in the seashell–mycelium composite, balancing the effect of mycelium growth in increasing dimensional change.

Although mycelium-bound composites are advantageous in terms of their mechanics and sustainable production methods. However, there are many limitations in adapting these mycelium-bound composites in mass production for constructional applications. Mainly, the patch-to-patch variations in physical and mechanical properties. Implying difficulty in standardizing their production method. Due to the non-standardized types of substrates for certain species of fungi to maximize the production yield of mycelium and to optimize the composite mechanics. In addition to the difficulty of standardizing the fungal species, since there are over one million species [127], with their varied microstructure and mechanics imposing difficulty of investigating the structure-mechanics relationship of different classes of fungi. Moreover, the lengthy production method of mycelium biocomposites due to the required a growth cycle duration, as well as the need for automatized controlling of the growth conditions, including temperature, humidity, substrate, light, and pH within an incubating environment with minimum human intervention.

For constructional applications, mycelium biocomposites including the currently proposed seashell–mycelium composite require further investigation of their acoustic absorption, thermal insulation, as well as pets and parasites resistance. Cellulose was used in the current study as an initial growth substrate; however, this might result in contamination with other fungal or bacterial strains or other parasites as insects [128]. This requires further investigation on customized fungal growth media that boost the growth of the specific desired strain while hindering other non-desirable ones.

## 4. Materials and Methods

### 4.1. Seashell Biocomposite Material Preparation

A seashell-based biocomposite was prepared from 120 g of Seashells Powder < 100 μm from the *Aequipecten* species, genus *Pectenas* and family *Pectinidae* [129]. It was used as an infill, reinforcement, source of calcium carbonate (CaCO_3_) and chitin. Obtained from local restaurants’ wastes. The seashells were washed with ethanol 70% and distilled water, respectively. Then, they were boiled for 30 min and dried in 250 °C for 20 min. After that, they were ground and sifted three times until they reached a homogenous powder of less than 100 µm particle size and of pH 8.6, mixed with 29 g of xanthan gum of pH 7 (C_35_H_49_O_29_ (monomer)) as a matrix. An amount of 1.44 g of acetic acid (CH_3_COOH) was used as a cross-linking agent of pH 3. An amount of 4 mL of glycerol (C_3_H_8_O_3_) was added as a plasticizer with a pH 7. Finally, 120 mL water was added as the solution media. The mixture of pH 7 was prepared in sterile conditions at room temperature immediately before the 3D printing process of the V3 linear brick model (V3-LBM) profiles.

### 4.2. 3D Printing by Paste Extrusion

A domestically modified Ender 3D printer for paste extrusion using air pressure piston was used to print the three profiles of the V3-LBM. The dimensions of the original V3-LBM design were resized to fit the printing bed dimensions of the 3D printer paste extruder to the following: 221 × 109 × 34 mm^3^ with a %75 scaling down ratio. The printing settings were adjusted to the following: layer height 1 mm, nozzle diameter 1.8 mm, tolerance/resolution 0.04, z height offset 1 mm, jog speed 10 mm/s, print speed 20 mm/s, extrusion multiplier 4, step back air travelling 2.5, extrusion wait 200 millisecond and pressure control activated. Three profiles of the V3-LBM were printed. Each was composed from 5 layers, with a final dimension for the profile of 221 × 109 × 5 mm^3^. The 3D printing-paste extrusion process was conducted in sterile conditions at room temperature. The 3D-printed profiles were left to dry for 48 h at room temperature, in sterile conditions.

### 4.3. P. ostreatus Culturing (Culture Medium and Conditions) and V3 Biocomposite Brick Inoculation (Seeding)

One roll of toilet paper was purchased from the local market. The sheet dimensions per roll is 13.8 × 10.8 cm, roll length 16.6 m, and roll weight 160 g. Each sheet is composed of 3 layers of 100% pure virgin cellulose (C_6_H_10_O_5_)n. The cellulose paper role was soaked in 1 L of boiling distilled water with stirring for 15–20 min until forming a gel-like solution. The solution was cooled down to 35 °C. Then, 50 mL of the cellulose solution was poured between every two profiles (50 mL between the bottom layer of the top profile and the top layer of the middle profile, and another 50 mL between the bottom layer of the middle profile and the top layer of the bottom profile) of the 3D-printed profiles of the V3-LBM in sterile conditions. Fresh King Oyster Mushrooms *Pleurotus ostreatus* were purchased from the local market at the same day of cultivation and washed with 30% ethanol then three times with distilled water. An amount of 100 g of the 1 cm^3^ mushroom pieces were then added between every two cellulose-painted profiles of the biocomposite 3D printed profiles of the V3-LBM. The cutting of the mushrooms was conducted using a sterile forceps and scalpel in sterile conditions at room temperature 25 °C. The three profiles coated with the cellulose gel and mushroom pieces in between them, were sealed together with a sterile plastic wrap and incubated in dim light of 3 footcandle of white light and 25 °C for 4 weeks. The development of the fungal culture and the mycelium were checked every week in sterile conditions.

### 4.4. SEM Microscopy and EDX Chemical Analysis (Sampling, Sample Preparation)

After four weeks of incubating the seashell–mycelium V3-LBM. Three samples from different zones and orientations were extracted from the brick using a sterile scalpel and forceps with a diameter of 1.5 cm per sample. The samples were taken from the top-left-side border labelled S, the middle zone labelled M, and the lateral zone of the biowelded brick labelled L. The samples were examined to detect the welding of every two profiles where the mycelium was grown; these are (between the bottom layer of the top profile and the top layer of the middle profile, and the bottom layer of the middle profile and the top layer of the bottom profile). The three samples were dried at 130 °C inside an oven for 90 min performed on two consecutive cycles with 12 h between each cycle. Then, they were preserved separately in sealed sterile containers at room temperature. The brick dimensional change was estimated, after the growth of *P. ostreatus* mycelium, and after drying the final seashell–mycelium brick. The samples were then prepared for the SEM by mounting them on metal stubs of 1 cm diameter using a sticky carbon disc. Then they were coated with the silver conductive paint to prevent charge buildup on specimen surface. After preparing the samples they were examined using the Scanning Electron Microscope and EDX chemical elements analysis at the University of Barcelona Facilities.

## Figures and Tables

**Figure 1 biomimetics-08-00504-f001:**
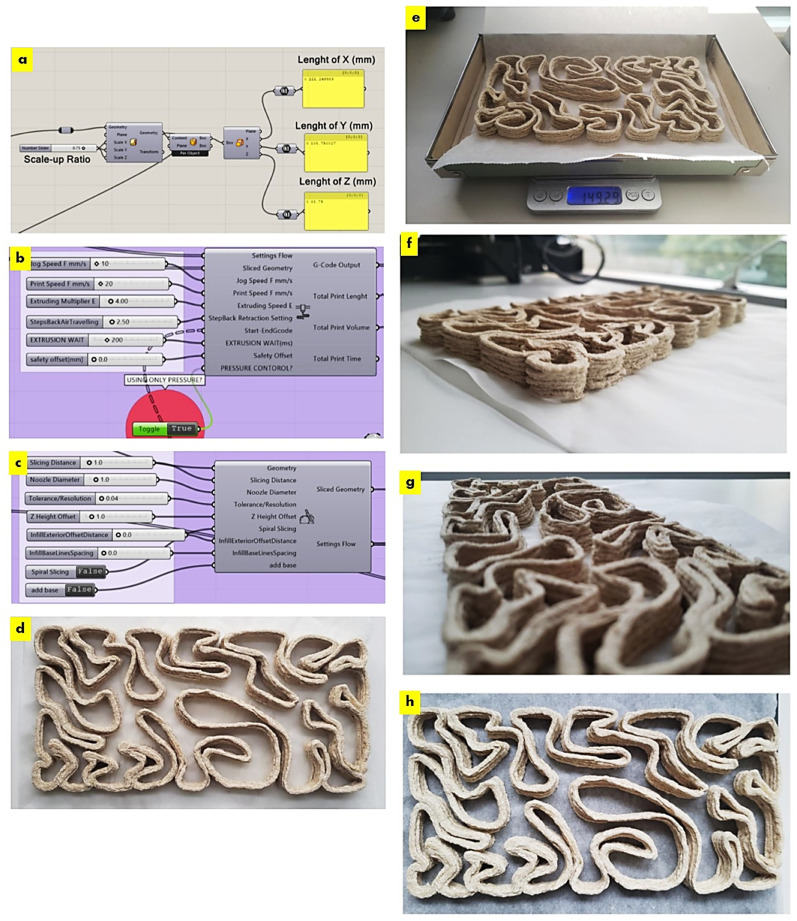
The 3D printing paste-extrusion settings and design customization by scaling down the original V3 linear brick model to fit the dimensions of the printing bed of the 3D printer paste extruder, reaching 221 × 109 × 34 mm. (**a–c**) exhibit the developed algorithm for the uniform parametric scaling of the V3-LBM. (**d**) the shape retention post printing per layer with mild swilling of 5 mm in all directions. (**e**) the final weight of the 3D-printed profile composed of 5-layer post-drying, which is 149.29 g. (**f**–**h**) t different views of the 3D printing tests for printing V3-LBM profiles; (**f**) pre-drying; (**g**) and (**h**) post-drying, showing the overall shape fidelity and layer to layer adhesion of the seashell-based biocomposite paste.

**Figure 2 biomimetics-08-00504-f002:**
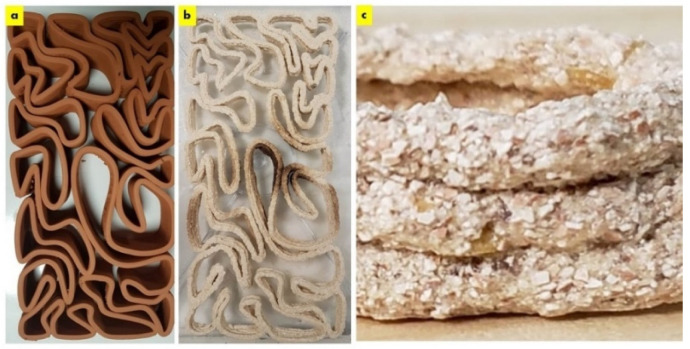
A comparison between the clay brick and the seashell biocomposite brick of the same V3-LBM. (**a**,**b**) show the textured rough finish of the seashell-based biocomposite, 3D-printed V3-LBM in comparison to its clay brick twin. (**c**) exhibits a 10× magnified image of seashell-based biocomposite material layer adhesion, showing the big particle size range from 100–500 µm.

**Figure 3 biomimetics-08-00504-f003:**
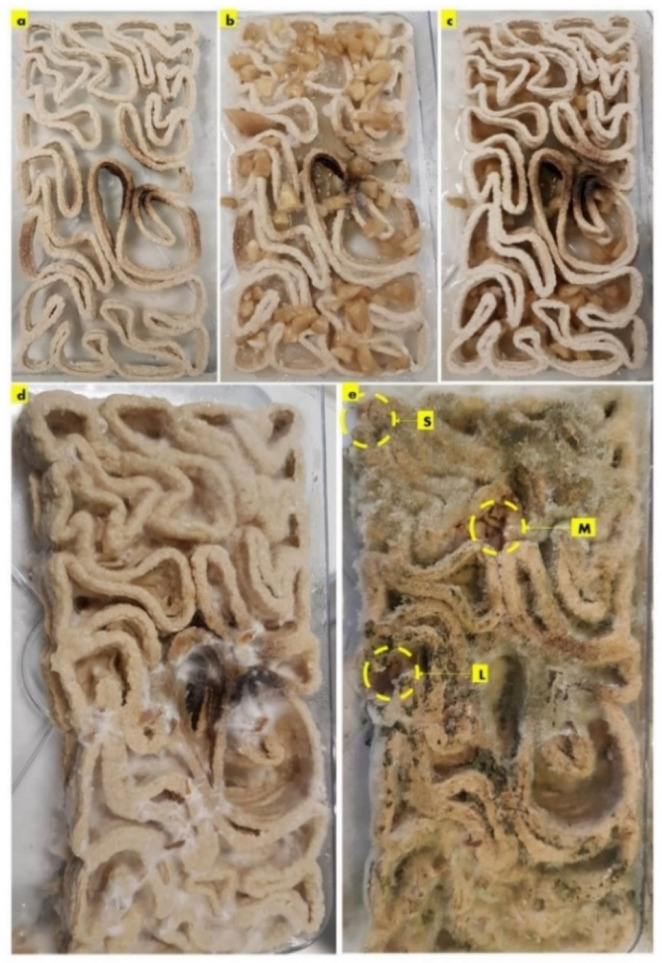
The *P. ostreatus* mycelium development on the 3D printed seashell biocomposite V3-LBM profiles, during four weeks of cultivation. (**a**–**c**) exhibit the seeding steps of the *P. ostreatus* mushroom pieces between the separate three profiles. (**d**) exhibits the fungal hyphae growth and development after one week of cultivating *P. ostreatus*, and (**e**) exhibits the four-weeks old fungal culture embedded and covering the entire biowelded brick, biowelding the three separate 3D printed profiles of the V3-LBM. It also exhibits the location where the three samples S, M, and L were extracted for the SEM-EDX analysis.

**Figure 4 biomimetics-08-00504-f004:**
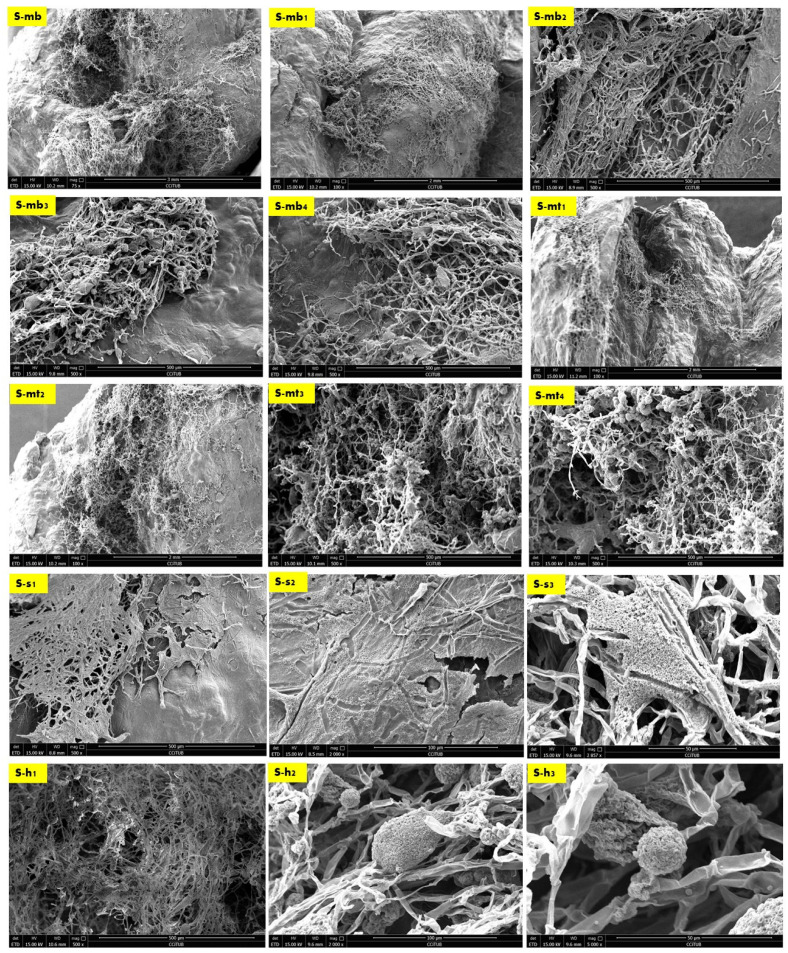
The SEM study of the biowelding effect by *P. ostreatus* mycelium and the welding of the separate 3D-printed seashell biocomposite profiles of the V3-LBM. The side sample S taken from the top-left side corner of the welded brick. The SEM images exhibit the hyphal network biowelding every two separate profiles from different orientations as follows: **S-mb** to **S-mb_2_** under magnifications of 75×, 100×, and 500×, respectively, detect the fungal hyphae bridging the two layers of the two separate profiles of the middle and the bottom. **S-mb_3_** and **S-mb_4_** show a closeup of these fungal hyphae bridges. Similarly, **S-mt_1_** and **S-mt_2_** exhibit overall views under 100× of the biowelded middle and top profiles showing the fungal hyphal bridges between both layers, while **S-mt_3_** and **S-mt_4_** exhibit the dense mycelium in these bridges under 500× magnification. **S-s_1_** to **S-s_3_** exhibit the surfaces of the biowelded layers between the middle and top profiles under 500×, 2000×, and 2857×, respectively. **S-h_1_** to **S-h_3_** focus on the dense hyphal network that composes the mycelium penetrating the seashell–biocomposite layers between the middle and top profiles of the 3D-printed biocomposite profiles**. S-h_1_** under 500× exhibits the overall dense mycelium, while **S-h_2_** and **S-h_3_** exhibit the fungal spores and hyphae under 2000× and 5000×, respectively.

**Figure 5 biomimetics-08-00504-f005:**
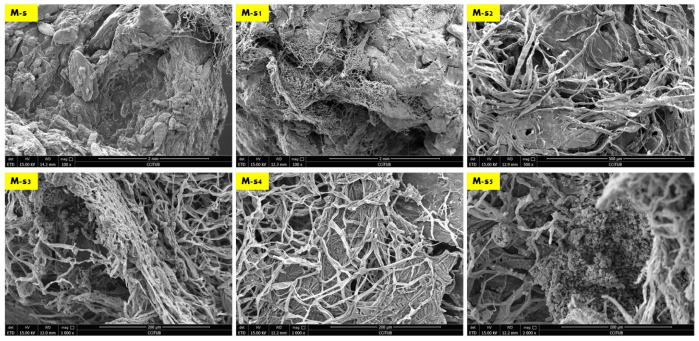
The SEM images of the M sample taken from the middle interior zone of the V3-LBM. **M-s** exhibits the biocomposite mass between the top and middle profiles with mild to low mycelium penetration on the upper side of the conjunction between their layers. **M-s_1_** and **M-s_2_** exhibit the biocomposite bottom layer of the top profile from its upper side surface that is biowelded with the top layer surface of the middle profile, under 100× and, 500×, respectively, with low fungal hyphae density due to the heterogenous surface extension and the curvature angle of the curvilinear form of this part of the V3-LBM. **M-s_3_**, and **M-s_4_** under 1000× each exhibit the bottom surface of the top profile (bottom layer) that is biowelded with the top layer of the middle profile, showing the significant increase in the density of the mycelium**. M-s_5_** exhibits the upper surface of the bottom profile biowelded with the bottom surface of the middle profile, under 1000×. It exhibits a moderate to high density of the mycelium.

**Figure 6 biomimetics-08-00504-f006:**
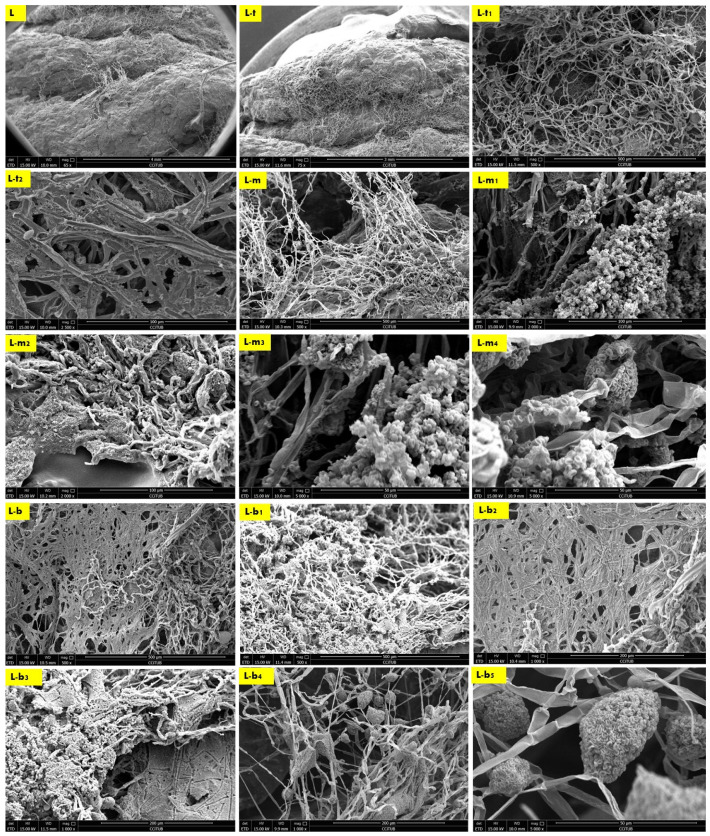
The SEM images of the L sample taken from the Lateral exterior zone of the V3-LBM. **L** under 65× and **L-t** under 75× show the dense mycelium bridging the two layers, the bottom layer from the top profile and the top layer from the middle profile. **L-t_1_** under 500× and **L-t_2_** under 2500× display the intertwining dense mycelium forming the bridges biowelding the bottom layer of the top profile and the top layer of the middle profile. **L-m** under 500× exhibits the dense mycelium bridging between the top layer of the middle profile and the bottom layer of the top profile. **L-m_1_**, and **L-m_2_** under 2000× show the dense fungal hyphae penetration to the biocomposite mass of the top layer of the middle profile. **L-m_3_** and **L-m_4_** exhibit details from **L-m_1_** and **L-m_2_**, respectively, under 5000× exhibiting the active colonization of the fungal spores and dense hyphal net. Similarly, **L-b**, **L-b_1_**, and **L-b_2_** under 500× and 1000×, respectively, represent the dense fungal hyphae between the bottom layer of the middle profile and the top layer of the bottom profile. **L-b_3_** under 1000× exhibits the integration and penetration of the fungal hyphae to the biocomposite mass binding the middle and bottom profile. Finally, **L-b_4_** under 1000× and **L-b_5_** under 5000× exhibit the fungal hyphae and spores.

**Figure 7 biomimetics-08-00504-f007:**
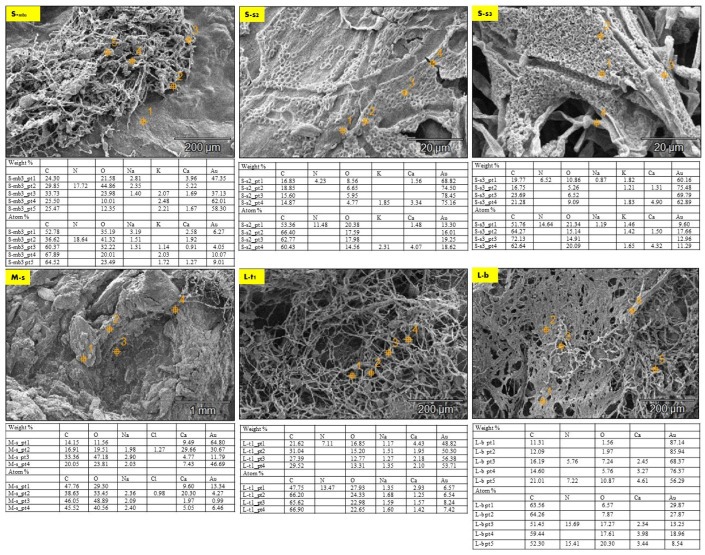
EDX chemical analysis of representative SEM images from the three samples (S, M, and L) detecting the chemical composition comparison between the seashell biocomposite and the biowelded biocomposite with mycelium as the binding agent between the separate 3D printed profiles of the V3 -LBM. SEM images were chosen to give an overall insight into the varied chemical composition between these zones (**S-mb_3_**, **S-s_2_**, **S-s_3_**, **M-s**, **L-t_1_**, and **L-b**). Each SEM image includes multiple points on the examined topology to analyze on a micrometer scale the distribution and variance of the chemical elements encompassed in the topology of each zone. The chemical elements included in each zone at different points are exhibited in a table below each image.

**Figure 8 biomimetics-08-00504-f008:**
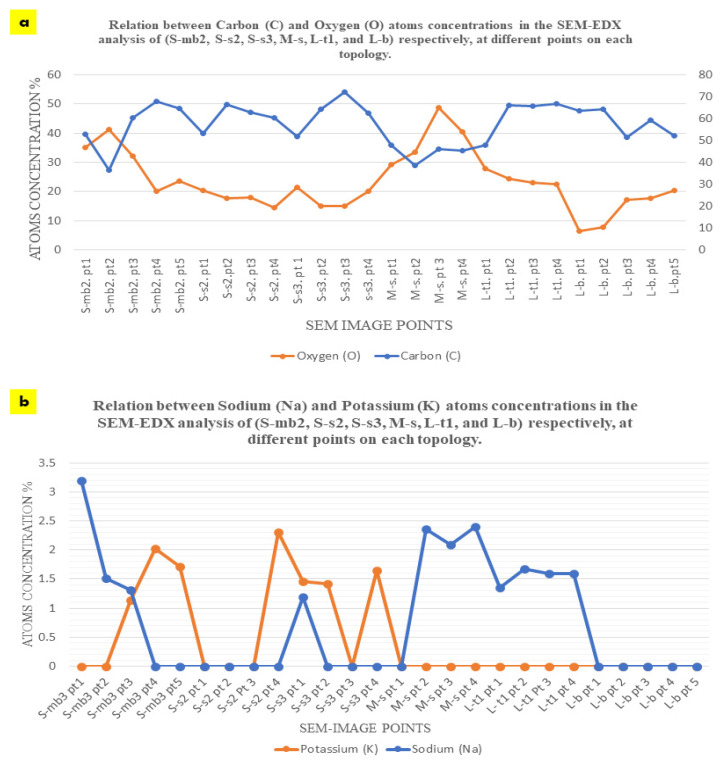
EDX-SEM analysis of the concentrations of four elements (carbon, oxygen, sodium, and potassium) on the vertical axis, in the examined SEM images (S-mb_3_, S-s_2_, S-s_3_, M-s, Lt_1_, and L-b) on the horizontal axis. (**a**) The relationship between carbon and oxygen atoms’ concentrations at different points of the examined zones, exhibiting the inverted relationship between carbon and oxygen, while showing the overall lower concentration of oxygen. (**b**) The relationship between sodium and potassium atoms’ concentrations in different points of the examined zones, exhibiting the inverted relationship between sodium and potassium, where sodium is present in areas that encompass more seashell biocomposite than fungal hyphae and potassium exists more in dense mycelium zones.

**Figure 9 biomimetics-08-00504-f009:**
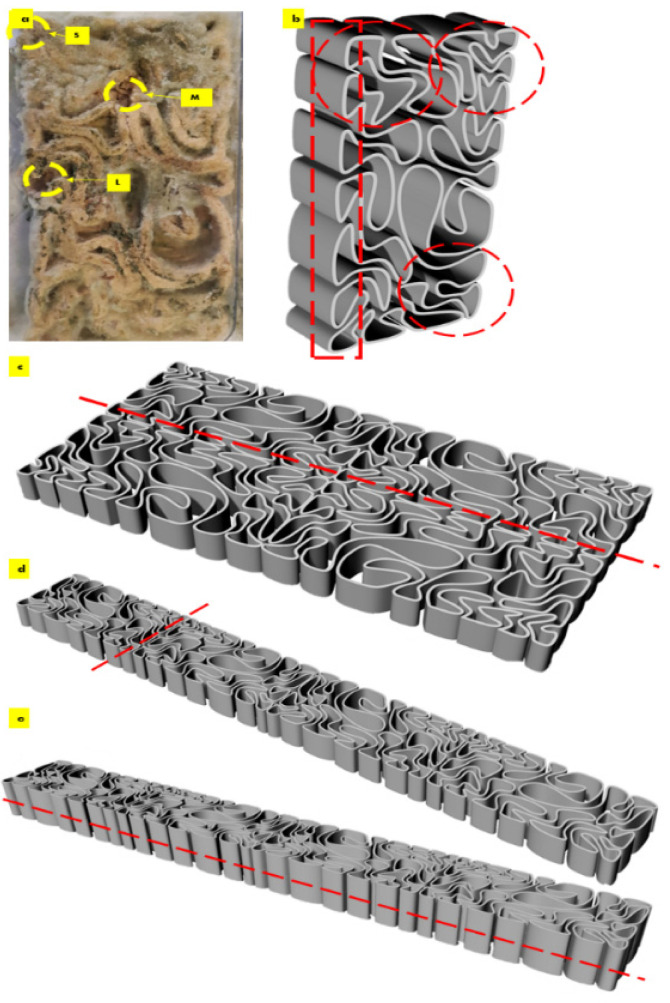
The staking and tessellation of the seashell–mycelium V3-LBM for enhanced biowelding effect. (**a**,**b**) Exhibit the best zones of biowelding effect according to the results of the SEM study, while (**c**,**d**) exhibit the favored method of tessellation of the seashell–mycelium V3-LBM by mirroring along the lateral side axis of the brick model that proved its geometry effect on increasing the contact points, hence increasing the biowelding effect. (**e**) exhibits the favorable staking method of the V3-LBM in the Z-direction where the bricks are placed exactly on top of each other without shifting. To maintain the extension of fungal hyphae penetrating the solid layers of the biocomposite and increasing the biowelding effect along the Z-axis.

## Data Availability

All the research data are presented in the manuscript; raw data are available upon request to corresponding author.

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
