# Peer review of "Biowelding 3D-Printed Biodigital Brick of Seashell-Based Biocomposite by Pleurotus ostreatus Mycelium"

_biomimetics, 2023, doi:10.3390/biomimetics8060504_

Round 1
Reviewer 1 Report
Dear Authors,
Thank you very much for providing us such a nice and interesting scientific work. Some minor changes should be forwarded to your work as follows:
1. lines 285-286 and 292 you are referring to dimension using mm3 while probably should be mm
2. line 300 probably hight should be changed to height
3. lines 298-307 the sentences should be rephrased as the explanation regarding weights is not clearly presented and reading enough times is a must to understand the point of this paragraph
4. in many cases in your study you are writing "...Figure X. ..." and then a capital letter follows while the sentence continues. Please make the necessary changes so as a reader follows the correct points in a sentence, examples were found in lines 309,353,355,356,503,692 etc
English language is generally in a very comprehensive and scientific level, minor fixes should be done
Author Response
The authors thank the reviewer for their constructive comments and here are the corresponding answers to each comment coupled with applying the required corrections in the manuscript where applicable:
Comment 1: Thank you very much for providing us with such a nice and interesting scientific work. Some minor changes should be forwarded to your work as follows: In lines 285-286 and 292 you are referring to dimension using mm3 while probably should be mm.
Answer 1: Done, the mm3 was corrected to mm in lines 285-286, and 292. the corrections are performed using the tracking system in Microsoft Word.
Comment 2: line 300 probably hight should be changed to height.
Answer 2: Done, the correction was made in line 300 replacing hight with height.
Comment 3. In lines 298-307 the sentences should be rephrased as the explanation regarding weights is not clearly presented and reading enough times is a must to understand the point of this paragraph.
Answer 3: Done from 298-307, the paragraph was rephrased to give a simpler and more direct explanation.
Comment 4: in many cases in your study you are writing "...Figure X. ..." and then a capital letter follows while the sentence continues. Please make the necessary changes so that a reader follows the correct points in a sentence, examples were found in lines 309,353,355,356,503,692 etc.
Answer 4: Done, all figure captions were corrected following the comment of the reviewer.
Comment 5: English language is generally in a very comprehensive and scientific level, minor fixes should be done.
Answer 5: a general English language, grammar, and style revision was performed all over the manuscript with corrections done where applicable.
The authors would like to sincerely thank the reviewer for thier constructive comments and kind support.
Reviewer 2 Report
This work is very interesting and novel. I have some observations that might improve its enjoying and benefit from the reader.
The Introduction is very long, but I have no problems with it. Only, the first section is supposed logically to be the last, as it deals with the biobricks.
I would move the section 1.1., excluding the last paragraph "In the following section", at the end of the Introduction and rearrange everything as the consequence of it.
I don't find other problems, one of the best works read recently.
Please check the English structure, some sentences are cut in an awkward way.
Author Response
The authors would like to thank the reviewer for thier constructive comments and kind support, here are the answers:
Comment 1: This work is very interesting and novel. I have some observations that might improve its enjoyment and benefit the reader. The Introduction is very long, but I have no problems with it. Only, the first section is supposed logically to be the last, as it deals with the biobricks. I would move section 1.1., excluding the last paragraph "In the following section", at the end of the Introduction, and rearrange everything as a consequence of it.
Answer 1: the authors would like to extend extensive thanks to the reviewer for thier kind and fair evaluation of the presented study in this manuscript.
Although the authors understand and agree with the reviewer with their suggestion of the logical order of the topics presented in the introduction section in relation to the sequence of the experimental procedures and the exhibited results. however, the authors intended to start from the last step in their previously published work about the Biodigital Clay Bricks and the V3 linear brick model that achieved elastic and compressive strength properties. The current study is following on to test further properties and potentials of the V3 linear brick model of the biodigital bricks as stated in the two previous publications that explained these bricks' form generation and simulation process and their mechanical properties. thus the current study emphasizes the bioreceptivity of this V3-LBM by using developed seashell-based biocomposite and employing mycelium as a binder agent for the 3D printed profiles of this brick model. To propose the combination of digital manufacturing and biomanufacturing as a fabrication method of bricks in an affordable sustainable way. Given that the V3-LBM is the protagonist of the study, the authors wanted to highlight this by presenting it first in the introduction section, while all that comes after it in the introduction section presents an explanation and a state-of-the-art of how to achieve the proposed objective.
Comment 2: I don't find other problems, one of the best works read recently.
Answer 2: the authors extend their sincerest thanks to the reviewer for their kind and constructive comments.